# Diversification of Bourbon Virus in New York State

**DOI:** 10.3390/microorganisms11061590

**Published:** 2023-06-15

**Authors:** Rachel E. Lange, Alan P. Dupuis, Alexander T. Ciota

**Affiliations:** 1Department of Biomedical Sciences, School of Public Health, University at Albany, State University of New York, 1 University Place, Rensselaer, NY 12144, USA; rlange@albany.edu; 2Wadsworth Center, New York State Department of Health, Griffin Laboratory, 5668 State Farm Road, Slingerlands, NY 12159, USA; alan.dupuis@health.ny.gov

**Keywords:** Bourbon virus, New York State, *Amblyomma americanum*

## Abstract

Bourbon virus (BRBV, family *Orthomyxoviridae*) is a tickborne virus recently detected in the United States (US). BRBV was first identified from a fatal human case in 2014 in Bourbon County, Kansas. Enhanced surveillance in Kansas and Missouri implicated *Amblyomma americanum* as the primary vector for BRBV. Historically, BRBV was only detected in the lower midwestern US, but since 2020 it has been reported in North Carolina, Virginia, New Jersey, and New York State (NYS). This study aimed to elucidate genetic and phenotypic characteristics of BRBV strains from NYS through whole genome sequencing and the assessment of replication kinetics in mammalian cultures and *A. americanum* nymphs. Sequence analysis revealed the existence of two divergent BRBV clades circulating in NYS. BRBV NY21-2143 is closely related to the midwestern BRBV strains but has unique substitutions in the glycoprotein. Two other NYS BRBV strains, BRBV NY21-1814 and BRBV NY21-2666, form a distinct clade unique from previously sequenced BRBV strains. Phenotypic diversification was also detected in NYS BRBV strains compared to each other and midwestern BRBV strains, with BRBV NY21-2143 displaying attenuation in rodent-derived cell culture and a fitness advantage in experimentally infected *A. americanum*. These data suggest genetic and phenotypic diversification of emergent BRBV strains circulating in NYS that could contribute to increased spread of BRBV in the northeastern US.

## 1. Introduction

Bourbon virus (BRBV), a negative-sense, segmented RNA virus, belongs to the *Orthomyxoviridae* family, which consists of seven genera: four genera containing influenza viruses, one genera of salmon-specific viruses, and two genera of arthropod-borne (arbo) viruses [1]. BRBV is a part of the Thogotovirus genus that includes tick-transmitted arboviruses, Dhori, Oz, Thogoto, and Upolu viruses [1,2,3,4]. These viruses are predominantly found in Asia, Africa, and the Middle East, while BRBV and Aransas Bay virus are the only identified thogotoviruses in North America. BRBV is a human pathogen while Aransas Bay virus primarily circulates in wildlife [5]. 

All arboviral thogotoviruses are maintained between invertebrate and mammalian hosts with differences in the primary vector matching the known geographic range [1]. Thogotoviruses outside of North America are maintained by *Hyalomma* and *Amblyomma* spp. ticks and some mosquito species [1,2,4]. *Amblyomma americanum* have been incriminated as the primary vector of BRBV after expanded surveillance and testing of multiple tick species (*Amblyomma* spp., *Dermacentor* spp., *Ixodes* spp., and *Haemaphysalis* spp.) in Kansas and Missouri [6,7]. Additionally, laboratory studies demonstrated the vector competence of *A. americanum* with vertical and cofeeding transmission observed following experimental infection of naïve ticks [8]. While BRBV has primarily been isolated from *A. americanum*, detection in *H. longicornus* in Virginia highlights potential roles for other tick genera in BRBV maintenance in nature [9]. A mammalian reservoir for BRBV has not been identified, but serological testing in Missouri and New York identified white-tailed deer (*Odocoileus virginianus*) and/or raccoons (*Procyon lotor*) as potential sentinels for tracking BRBV spread [10,11]. Human sera testing has also determined that ~1% of people living in endemic regions of Missouri have neutralizing antibodies against BRBV [12].

The first human case of BRBV was identified in 2014 from Bourbon County, Kansas [13]. Nonspecific symptoms, including severe febrile illness following a reported tick bite, led to hospitalization of the patient. Eleven days after symptom onset, the patient died. Since 2014, four additional human cases have been identified in the midwestern United States (US) [13,14,15,16]. Symptoms in each case were associated with tick exposure and onset of symptoms including weakness, fatigue, and nausea 2–7 days post reported tick bite. Leukopenia and thrombocytopenia were observed in all three fatal cases of BRBV and now act as potential identifiers of BRBV infection in humans [1,13]. Fatal cases are associated with shock, organ failure, cardiac dysregulation, and acute bone marrow suppression [13,16]. BRBV detection in *A. americanum* and human cases also overlap with the geographic range of Heartland virus (family *Phenuiviridae*), which similarly presents as nonspecific flu-like symptoms after tick exposure, making clinical diagnosis and treatment difficult [17,18]. The emergence of a human pathogenic thogotovirus has prompted investigation of antivirals and vaccines against BRBV, but no specific preventative or therapeutic interventions are currently available [15,19].

While human cases have only been identified in the midwestern US, BRBV has been detected in *A. americanum* ticks across the midwestern, southeastern, and northeastern US. Similar to *Ixodes* spp., *Amblyomma* spp.’s geographic range has increased northward as a result of climate change and host expansion [20,21]. Historically, BRBV was only detected in Missouri, Kansas, and Oklahoma, but since 2020, BRBV has been reported in North Carolina, Virginia, New Jersey, and New York State (NYS) [9,11,22,23]. Despite these recent detections, the characterization of BRBV strains has been limited to midwestern isolates [24,25]. Therefore, we aimed to assess the genetic and phenotypic diversity of the first BRBV isolates from NYS through whole genome sequencing and phenotypic characterization in vitro and in vivo. Our results demonstrate important strain and host-specific differences that inform our understanding of BRBV evolution and transmission potential.

## 2. Materials and Methods

### 2.1. Viruses

All work was conducted in a biosafety level 3 facility at the New York State Department of Health (NYSDOH). Viral positives were identified through the NYSDOH tick surveillance and testing programs [11]. BRBV-positive tick pools were identified using a quantitative reverse transcriptase polymerase chain reaction (qRT-PCR) assay as described previously [11]. All NYS BRBV positive tick pools were *A. americanum* nymphs collected in Suffolk County, NYS, in 2021. Viral isolates were obtained by amplification of positive tick homogenates on African green monkey kidney cells (Vero, ATCC, CCL-81). Virus positive cultures were confirmed by qRT-PCR, harvested, clarified, and supplemented with heat-inactivated fetal bovine serum (FBS, Hyclone, Logan, UT, USA).

### 2.2. Sequencing

RNA was extracted from qRT-PCR positive tick pool homogenates using an automated MagMAX nucleic acid extraction kit and associated instrument (ThermoFisher Scientific, Waltham, MA, USA). Full genome sequences were obtained by amplification of each segment (Appendix A) using a Superscript III One-Step RT-PCR kit (ThermoFisher Scientific, Waltham, MA, USA). Segment amplicons were pooled for next generation sequencing (NGS) carried out on an Illumina MiSeq platform (2 × 500 bp PE) (Illumina, San Diego, CA, USA) at the Wadsworth Center Advanced Genomics Technologies Core.

Reads were paired, merged, and mapped to a BRBV reference genome (Bourbon virus strain original: KU708248-55) in Geneious Prime 2021.2 to obtain consensus sequences. Individual amino acid substitutions were identified based on MAFFT alignments of full genome consensus sequences for each segment compared to available sequences from Genbank (Appendix A) [26]. Phylogenetic trees were generated based on this MAFFT alignment using the Geneious Tree Builder function for neighbor joining trees, Tamura-Nei distances, and 1000 bootstraps. Trees were visualized in iTOL Interactive Tree of Life v6.

### 2.3. In Vitro Growth Kinetics

African green monkey kidney cells (Vero, ATCC, CCL-81), baby hamster kidney cells (BHK-21, ATCC, CCL-10), and human hepatocellular carcinoma cells (Huh7, ATCC) were grown in minimal essential media (MEM, Gibco, Invitrogen Corp, Carlsbad, CA, USA) with heat-inactivated fetal bovine serum (FBS, Hyclone, Logan, UT, USA) and maintained at 37 °C, 5% CO_2_. Confluent monolayers were generated by seeding six-well plates with the respective cell type and were maintained for 3–4 days prior to experimental infection to reach confluency (1 × 10^6^ cells per well).

Viral growth kinetics in each cell line were carried out as previously described [27]. Briefly, six-well tissue culture plates of confluent monolayers of each respective cell type were inoculated in triplicate with a multiplicity of infection (MOI) of 0.01 plaque-forming units (PFU) per cell. Inoculum was allowed to adsorb for 1 h at 37 °C, 5% CO_2_ followed by washing the wells with 2 mL of phosphate-buffered saline (PBS) three times. Maintenance media was added, and 0.1 mL of supernatant was collected at 24-h intervals up to 120 h post-infection (HPI). Samples were stored at −70 °C in a 1:10 dilution of BA-1 media (M199 medium with Hank’s salts, 1% bovine albumin, TRIS base (tris [hydroxymethyl] aminomethane, sodium bicarbonate, 20% FBS, and antibiotics) until processing.

Viral load was determined by plaque assay on Vero cells. Confluent monolayers were inoculated with 0.1 mL of sample in duplicate and were allowed to adsorb for 1 h at 37 °C, 5% CO_2_. An overlay of MEM, 10% FBS, and 0.6% oxoid agar was added to each well after adsorption and allowed to incubate for 5 days before the addition of a second overlay containing MEM, 2% FBS, 0.6% oxoid agar, and 2% neutral red (Sigma-Aldrich Co. St. Louis, MO, USA). After an overnight incubation at 37 °C, 5% CO_2_, plaques were counted, and viral titers were determined as PFU/mL. Replication curves were generated based on mean titer and were analyzed using paired *t*-tests. All analyses and visualizations were carried out in GraphPad Prism version 9.0.1.

### 2.4. Synchronous Infection of Ticks and Viral Detection

The following reagent was provided by the Centers for Disease Control and Prevention for distribution by Biodefense and Emerging Infections (BEI) Resources, National Institutes of Allergy and Infectious Diseases (NIAID), National Institutes of Health (NIH): *Amblyomma americanum* Nymph (Live, NR-44124). Ticks were maintained at 95% relative humidity (RH), 20 °C, and in a 16:8 light:dark cycle. Whole nymphal ticks were infected by immersion as previously described [27,28,29]. Briefly, nymphs were held at reduced RH (40–65%) for 48 h prior to infection, followed by suspension in a 1 × 10^6^ dilution of each respective strain. Ticks were immersed for 1 h at 34 °C, washed with PBS twice, and dried individually before storage in Plaster of Paris jars. At 7-, 14-, 21-, and 28-days post-infection (DPI), whole ticks were moved to individual tubes with 5 mm stainless steel BBs (Daisy Outdoor Products, Roger, AR, USA) and were stored at −70 °C until processing.

For processing, whole nymphs were mechanically homogenized in 0.6 mL of diluent (20% FBS in Dulbecco’s PBS, 50 ug/mL penicillin/streptomycin, 5 ug/mL gentamicin, and 2 ug/mL of fungizone (Sigma-Aldrich, St. Louis, MO, USA)) using a Retsch Mixer Mill, MM 301 (Retsch Inc., Newtown, PA, USA) at 30 cycles/s for 4 min. RNA was then extracted from tick homogenates using the MagMAX nucleic acid extraction kit and was tested for BRBV using the qRT-PCR assay described above [11]. Viral load was determined based on a standard curve of ten-fold dilutions of a known BRBV viral stock and was expressed as a relative PFU/mL. Viral loads were compared by nonparametric one-way ANOVA with Friedman’s multiple comparisons post-hoc tests and infection rates were compared using Chi-squared tests conducted in GraphPad Prism version 9.0.1.

## 3. Results

### 3.1. NYS BRBV Strains Are Genetically Distinct from Midwestern US BRBV Strains

To determine the genetic profile of recently isolated NYS BRBV strains—BRBV NY21-1814, BRBV NY21-2143, and BRBV NY21-2666—full genome sequencing was conducted and consensus sequences of each segment were compared to available midwestern BRBV isolates from the US and representative Dhori-like and Thogoto-like strains (Figure 1). Compared to other Dhori and Thogoto viruses, all US BRBV strains cluster together and share 93–99% similarity across all segments at the nucleotide level (Table 1). Despite close genetic similarity, BRBV NY21-1814 and BRBV NY21-2666 cluster separately from the midwestern US BRBV strains and BRBV NY21-2143 in segments 1–4. This separation is a result of a ~6% difference at the nucleotide level, while segment 6 (matrix protein) only differs by ~3%. Synonymous mutations account for the majority of the genetic variability in all segments. Synonymous mutations shared between BRBV NY21-1814 and BRBV NY21-2666 were primarily located in segment 1 (PB2) and segment 3 (PA), while BRBV NY21-2143 displayed the largest number of mutations in segment 4 (GP) relative to midwestern BRBV strains. Shared synonymous mutations between all NYS BRBV strains were found in segment 1 (PB2) and segment 3 (PA). The highest percentage of variable nucleotides (6.9%) was identified in segment 3.

BRBV NY21-1814 and BRBV NY21-2666 share 33 amino acid substitutions which are distinct from all other US BRBV strains, including BRBV NY21-2143 (Table 2). These substitutions are primarily concentrated in segment 3 (PA), yet some variability exists in all segments. Although BRBV NY21-1814 and 2666 are genetically similar, they contain three and four unique substitutions, respectively. BRBV NY21-2143 has one unique substitution in segment 3 (PA) and two substitutions in segment 4 (GP). The only amino acid substitutions shared between all three NYS BRBV strains are found in segment 3 (PA- I224V and M636I). Notably, segment 5 (NP) is highly conserved across all US BRBV strains except for a single amino acid substitution shared between BRBV NY21-1814 and BRBV NY21-2666 (S300N).

### 3.2. NYS BRBV Strains Display Phenotypic Variability in Mammalian Cells

Viral replication kinetics were determined in distinct mammalian cell lines to better understand phenotypic differences associated with unique genotypes of NYS BRBV strains. BRBV Original was used as a representative strain from the midwestern US BRBV isolates (Appendix A). Growth kinetics were assessed in baby hamster kidney (BHK-21), African green monkey kidney (Vero), and human hepatoma (Huh7) cell lines to recapitulate maintenance in small mammalian hosts associated with the tick life cycle (BHK-21 and Vero) and human hosts (Huh7). Kinetics in BHK-21 were significantly different across all BRBV strains assessed (Figure 2A). BRBV Original replicated to the highest levels, followed by BRBV NY21-1814 (Figure 2A, Paired *t*-test, * *p* = 0.0314). Of note, BRBV NY21-2143 displayed significant attenuation relative to BRBV Original and BRBV NY21-1814 (Figure 2A, Paired *t*-test, ** *p* = 0.0054 and ** *p* = 0.0072, respectively). However, strain-specific growth kinetics were unique in Vero and Huh7 cultures, as a significant growth deficit was detected for BRBV NY21-1814 relative to BRBV Original and BRBV NY21-2143 (Figure 2B,C, Paired *t*-test, Vero *** *p* = 0.0004 and **** *p* < 0.0001, Huh7 * *p* = 0.0250 and ** *p* = 0.0044, respectively).

### 3.3. Fitness Advantage of NYS BRBV Strain BRBV NY21-2143 in Experimentally Infected Amblyomma americanum

To investigate the influence of BRBV strain on phenotypic variability in invertebrate hosts, nymphal *A. americanum* were infected by immersion and collected at 7-, 14-, 21-, and 28-days post-infection (DPI). Combined infection rates from all timepoints (~60%) were not significantly different (Chi-squared test, *p* = 0.09). Despite this, strain had a significant effect on overall viral load (nonparametric one-way ANOVA, * *p* = 0.0417). BRBV NY21-2143 grew to significantly higher viral loads than BRBV NY21-1814 (Figure 3, nonparametric one-way ANOVA with Friedman’s multiple comparisons post-hoc test, * *p* = 0.0400). Lower viral loads were also measured for BRBV Original relative to BRBV NY21-2143. While not statistically significant overall (*p* = 0.23), BRBV NY21-2143 viral load was significantly higher than that of BRBV Original at 28 DPI (*t*-test, * *p* = 0.039). These data indicate that BRBV NY21-2143 has an overall fitness advantage in nymphal *A. americanum* compared to midwestern and NYS BRBV strains.

## 4. Discussion

This study reported the genotypic and phenotypic characteristics of BRBV strains isolated in the northeastern US. Whole genome sequencing revealed the existence of two distinct clades in NYS, including a novel genotype divergent from previously sequenced midwestern strains. Previous studies indicate that tickborne viruses generally evolve slowly, suggesting this genetic separation is indicative of both distinct historical foci of BRBV and separate introductions into NYS [30]. While BRBV is a recently recognized thogotovirus, it has likely been circulating outside of the midwestern US undetected. However, BRBV may have recently spread to the northeast due to changing environmental conditions that created more suitable habitats for the primary vector *A. americanum*, sentinel mammals such as white-tailed deer, and anthropogenic land use changes that have altered biodiversity in this region [31,32,33]. *A. americanum* populations are already established in southern NYS and climate suitability models have predicted a continued expansion northward [21,34,35]. Additionally, white-tailed deer can support all life stages of *A. americanum* and have seen drastic increases in population density compared to historic counts [33,36,37].

The introduction of BRBV into NYS may have been facilitated by the dispersal of infected immature stages of *A. americanum* by birds during migration or post-breeding dispersal. Common songbird species including the Northern Cardinal (*Cardinalis cardinalis*), American Robin (*Turdus migratorius*), Song Sparrow (*Melospiza melodia*), Hermit Thrush (*Catharus guttatus*), and Gray Catbird (*Dumetella carolinensis*) span the midwestern US and southern NYS and are known to be infested with a diverse range of ectoparasites including *A. americanum*, *I. scapularis*, and *D. variabilis* [38]. This hypothesis is similar to the proposed spread of thogotoviruses from Africa to Europe [39]. The migration and long-distance movement of birds have been implicated in the expansion of other tickborne pathogens, primarily *Borrelia burgdorferi,* throughout the US and Canada [40,41,42,43,44].

The patterns of genetic variation in NYS BRBV genomes suggest segment-specific evolutionary pressures. Segment 3 (PA) is the most genetically diverse, followed by segments 1 (PB2) and 2 (PB1), two other components of the viral polymerase complex. The diversity of RNA viruses, which can be important for maintenance during host cycling, is largely facilitated by error-prone RNA-dependent RNA polymerases (RdRps), high replication rates, and gene-specific purifying selection [45]. Unlike the variability observed in the polymerase complex, both segments 5 (NP) and 6 (M) are highly conserved across all BRBV strains, despite geographic origin. This is consistent with previous studies that found that both segments act as immune antagonists under strict pressures within mammalian hosts [46,47,48,49,50].

Phenotypic diversity was observed between all NYS BRBV strains as compared to a representative midwestern strain in both in vitro and in vivo systems. Despite all three cell lines being interferon-incompetent (IFN), distinct, strain-specific fitness was observed [51,52,53]. Of note, growth kinetics were similar between the primate-derived cell lines Vero and Huh7, but distinct from the rodent-derived cell line BHK-21. Inherent differences between primates and rodents, including host codon usage, host receptor structures, and gene expression patterns, could contribute to the variability observed [54,55,56]. While BRBV NY21-1814 displayed significant attenuation in Vero and Huh7 compared to BRBV NY21-2143 and BRBV Original, it exhibited an intermediate phenotype in BHK-21. BRBV NY21-2143 grew to similar viral loads as BRBV Original in Vero and Huh7 but showed a significant growth deficit in BHK-21. This difference further supports host-dependent mechanisms leading to differing phenotypes of BRBV NYS strains. Replicative fitness in experimentally-infected *A. americanum* was also distinct among BRBV strains. BRBV NY21-2143 displayed a fitness advantage compared to BRBV Original and BRBV NY21-1814. Given the attenuation of this strain in BHK-21 cells, these results are consistent with an adaptive trade-off often noted in other arboviral systems [57,58]. To better understand strain-specific fitness, virulence, and the role of distinct substitutions in viral fitness, assessment in an animal model is needed. Animal models could further reveal strain-specific pathogenesis in mammalian hosts and be utilized to assess transmission from tick hosts. This, however, is challenging given that there are currently no established animal models for BRBV [15,24].

Despite BRBV NY21-1814 and BRBV NY21-2666 displaying the most genetic divergence, BRBV NY21-2143 was the most phenotypically distinct compared to other NYS and midwestern BRBV strains, with a majority of substitutions identified in segment 4 (GP). BRBV GP is unique in that thogoto- (THOV) and dhori-like virus (DHOV) GPs are highly homologous to the insect-specific baculovirus glycoprotein 64 (gp64) as compared to the envelope GPs of influenza viruses [59,60]. Baculovirus gp64 plays important roles in insect host cell entry and host range determination [61,62,63]. The unique GP substitutions identified in BRBV NY21-2143 could be responsible for the attenuation exhibited in rodent cell lines and fitness advantage in *A. americanum*. The GP is important for early viral processes such as host cell attachment, entry, and release into the host cell, which could contribute to this observed phenotypic variation. To further elucidate the role of these GP substitutions in NYS BRBV strain- and host-specific fitness, the development of novel genetic tools is needed. Currently, these tools are limited to reporter systems, and previous work is confined to THOV GPs and structural comparisons of orthomyxovirus GPs [19,59,64]. These studies, however, provide important insights into the potential consequences of these substitutions as THOV GPs have been shown to be physiologically active at tick pH (6–6.5) compared to mammalian pH (human-7.3) [65]. Additionally, when compared to known GP structures of THOV, DHOV, midwestern BRBV, and baculoviruses, the unique BRBV NY21-2143 GP substitutions reside in a domain that is highly sensitive to pH-dependent conformational changes [60]. While the BRBV NY21-2143 substitutions do not align with important cysteine or glycosylation sites identified in these other orthomyxoviruses, there is potential for these substitutions to contribute to the fitness tradeoff observed in experimental mammalian and invertebrate systems [59,60,64,65].

Overall, these studies highlight the need to maintain robust tickborne disease surveillance programs across states in this region. NYS has a widespread collection and testing program that facilitated the detection of BRBV; however, increased surveillance in other states with established *A. americanum* populations, in concert with genetic and phenotypic characterization, is vital for tracking viral spread and diversity. Additionally, the vector competence of other tick species present in the northeast for BRBV is unknown and spillover events into new transmission cycles could further expand BRBV establishment. These studies emphasize the importance of educational outreach to practitioners and the public to spread awareness of human disease symptoms and individual preventative measures when spending time outdoors.

## Figures and Tables

**Figure 1 microorganisms-11-01590-f001:**
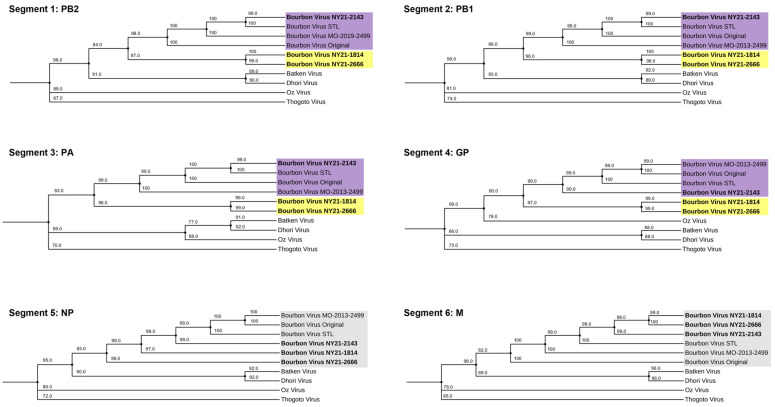
Maximum likelihood phylogeny of Bourbon virus (BRBV) and related Orthomyxoviruses based on the coding region of segments 1–6. Bootstrap values are displayed at each branchpoint (range: 65–100). Recent BRBV strains from New York State (NYS) cluster separately (yellow) from midwestern BRBV strains (purple) with the exception of BRBV NY21-2143 in segments 1–4. All BRBV strains cluster together in segments 5 and 6 (gray).

**Figure 2 microorganisms-11-01590-f002:**
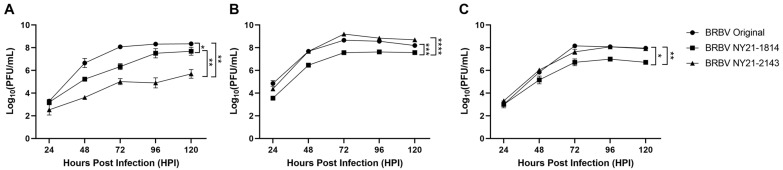
Growth kinetics of recent Bourbon virus (BRBV) strains from New York State (NYS) compared to the midwestern BRBV Original strain in baby hamster kidney ((**A**) BHK-21), African green monkey kidney ((**B**) Vero), and human hepatoma cells ((**C**) Huh7). Data points represent mean +/− SEM (n = 3 per strain). NYS BRBV strains showed variable kinetics across cell types. BRBV NY21-2143 showed significant attenuation compared to other BRBV strains in BHK-21 ((**A**) paired *t*-test, BRBV Original ** *p* = 0.0054 and BRBV NY21-1814 ** *p* = 0.0072). There were also significant differences between BRBV Original and BRBV NY21-1814 viral loads in BHK-21 ((**A**) paired *t*-test, * *p* = 0.0314). Replicative fitness displayed similar trends in Vero and Huh7 with BRBV Original and BRBV NY21-2143 exhibiting significantly higher viral loads compared to BRBV NY21-1814 ((**B**) Vero, paired *t*-test, *** *p* = 0.0004 and **** *p* < 0.0001, (**C**) Huh7, * *p* = 0.0250 and ** *p* =0.0044, respectively).

**Figure 3 microorganisms-11-01590-f003:**
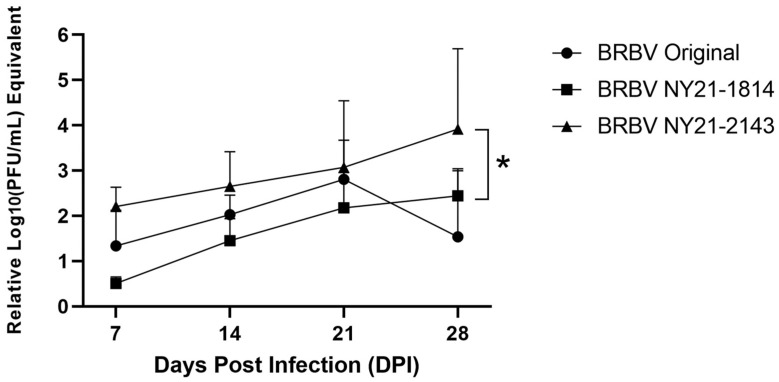
Growth kinetics of recent Bourbon virus (BRBV) strains from New York State (NYS) relative to the midwestern BRBV Original strain in experimentally infected *Amblyomma americanum* following immersion. Overall, BRBV NY21-2143 grew to significantly higher viral loads relative to BRBV NY21-1814 (Nonparametric one-way ANOVA with Friedman’s multiple comparisons post-hoc test, * *p* = 0.0400). BRBV NY21-2143 viral loads were not significantly higher overall relative to BRBV Original but were significantly higher at 28 days post-infection (DPI) (*t*-test, * *p* = 0.039).

**Table 1 microorganisms-11-01590-t001:** Nucleotide identity of Bourbon virus (BRBV) strains isolated in New York State (NYS) compared to midwestern Bourbon virus strains: BRBV Original, BRBV MO2013-2499, and BRBV-STL based on coding sequence. BRBV NY21-2143 shows the highest sequence identity to midwestern BRBV strains.

BRBV-1814			
Segment	BRBV-Original	BRBV-MO2013	BRBV-STL
1 (PB2)	94.83%	94.85%	94.85%
2 (PB1)	95.67%	94.44%	96.02%
3 (PA)	93.56%	93.77%	93.83%
4 (GP)	94.88%	95.08%	95.32%
5 (NP)	94.43%	94.65%	94.72%
6 (M)	97.91%	97.91%	97.79%
**BRBV-2143**			
1 (PB2)	99.17%	99.12%	99.34%
2 (PB1)	99.07%	97.31%	99.57%
3 (PA)	99.06%	98.85%	99.42%
4 (GP)	98.54%	98.47%	98.99%
5 (NP)	99.21%	99.49%	99.49%
6 (M)	99.69%	99.69%	99.57%
**BRBV-2666**			
1 (PB2)	94.79%	94.81%	94.81%
2 (PB1)	95.55%	94.33%	95.90%
3 (PA)	93.38%	93.59%	93.64%
4 (GP)	94.95%	94.88%	95.40%
5 (NP)	94.87%	95.08%	95.16%
6 (M)	97.91%	97.91%	97.79%

**Table 2 microorganisms-11-01590-t002:** Novel substitutions identified in recent Bourbon virus (BRBV) strains isolated in New York State (NYS) based on coding sequence compared to midwestern BRBV strains BRBV-Original, BRBV-MO2013, and BRBV-STL. Amino acid substitutions present in one strain and shared across multiple strains.BRBV NY21-1814 and BRBV NY21-2666 share many unique substitutions (n = 33) while BRBV NY21-2143 shares only two substitutions with the other NYS strains (**bolded**). Most substitutions were detected in segment 3 (PA).

Segment	Isolate	NT Change	AA Change
PB2	1814	A1125G	I375M
PB1	2666	A1069G	S357G
PA	1814	A1152G	I384M
PA	1814	A1439G	K480R
PA	2143	A893G	Q298R
PA	2666	C413A	P138Q
PA	2666	A1333G	I445V
GP	2143	A190G	I64V
GP	2143	A851G, T852C	N284S
GP	2666	A1379G	K460R
PB2	1814, 2666	A32G	K11R
PB2	1814, 2666	A1032G	I344M
PB2	1814, 2666	G1247A	R416K
PB2	1814, 2666	C1358T	A453V
PB2	1814, 2666	T1359G	A453V
PB2	1814, 2666	G1406A	S469N
PB1	1814, 2666	G167A	R56K
PB1	1814, 2666	A278G	H93R
PB1	1814, 2666	A1669T	I557L
PA	1814, 2666	A70G	I24V
PA	1814, 2666	G214A	V72I
PA	1814, 2666	G493A	A165T
PA	1814, 2666	A538G	T180A
PA	1814, 2666	A547G	T183A
**PA**	**1814, 2143, 2666**	**A670G**	**I224V**
PA	1814, 2666	G677A	R226K
PA	1814, 2666	A809G	H270R
PA	1814, 2666	A916G	I306V
PA	1814, 2666	A939C	K313N
PA	1814, 2666	G956A	R319K
PA	1814, 2666	A957G	R319K
PA	1814, 2666	T1086A	D362E
PA	1814, 2666	G1378A	V460I
PA	1814, 2666	A1397G	E466G
PA	1814, 2666	T1426C	F476L
PA	1814, 2666	A1445C	N482T
**PA**	**1814, 2143, 2666**	**G1908A**	**M636I**
GP	1814, 2666	C29T	A10V
GP	1814, 2666	T43G/C	S15P
GP	1814, 2666	A304G	T102A
GP	1814, 2666	A1219G	I407V
GP	1814, 2666	A1259G	L420R
NP	1814, 2666	G899A	S300N
M	1814, 2666	A428G	N143S
M	1814, 2666	A647G	K216R

## Data Availability

Data is contained within the article and Appendix A. Sequence data is publically available in NCBI GenBank [accession numbers can be found in Appendix A].

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
