# Peer review of "Diversification of Bourbon Virus in New York State"

_microorganisms, 2023, doi:10.3390/microorganisms11061590_

Round 1
Reviewer 1 Report
This study provides evidence that some strains of Bourbon virus isolated in New York are divergent from isolates obtained from the Midwest where the virus was first detected. The authors also report strain-specific differences in growth kinetics in mammalian cell culture lines and in Amblyomma ticks, although it is not yet possible to determine if these differences affect virulence. Several sections of the manuscript require clarification.
1. Lines 52-55: some of the references appear to be "frameshifted"; eg, the reference for neutralizing antibodies in Missouri should be reference 13, not 12, and the first case of BRBV is described in reference 14, not 13.
2. Figure 1 needs to include scale bars showing the extent of nucleotide values for each gene segment. Bootstrap values also need to be shown, at least at the branchpoints demonstrating that two of the NY isolates form a distinct clade.
3. In Table 2 it is not clear what the comparator is for the columns labeled "NT change" and "AA change." I assume it is BRBV-original, but this should be stated.
4. Lines 298-305 are confusing since they state that NY21-2143 is the divergent strain whereas, at the genetic level, Figure 1, Table 1, and Table 2 demonstrate indicate that NY21-1814 and NY21-2666 represent the divergent clade. Please clarify.
Reviewer 2 Report
Line51: Procyon lotor not iotor
Line 80: Materials & Methods, there is no mention of whether the work was undertaken in a BSL-3 or BSL-4 laboratory. Given the potentially deadly nature of the viruses studied, one would think that this question would be included/disclosed in the Materials & Methods section.
Line 133: Use of acronyms. Although it is, perhaps, obvious, ought not NIH, and NIAID be written out and then used as acronyms? Also, is BEI some type of an acronym? If so, desirable to spell out what this stands for.
Line 149: PFU is, presumably acronym for Plaque-forming unit? Again, ought be 'spelled out' first.
Lines 264-266: The recent and prodigious work of Scott and colleagues may be worthy of mention and inclusion regarding role of birds in spread of tick-borne infections.'
Scott JD, Anderson JF, Durden LA. Widespread dispersal of Borrelia burgdorferi-infected ticks collected from songbirds across Canada. J Parasitol. 2012 Feb;98(1):49-59. doi: 10.1645/GE-2874.1. Epub 2011 Aug 24. PMID: 21864130.
The work described in the manuscript is important, and also, frightening. The 'assumption' that Bourbon virus and its disease in humans is comfortably a 'mid-West' problem, is obviously destroyed by these findings. However, better to be aware, forewarned and 'forearmed'. However, still disturbing is that effective treatment for established infections in humans/mammals is not available. So, there are even 'worse' things in ticks than (treatable) Lyme disease, babesiosis, ehrlichiosis & anaplasmosis. There have been, it seems, increasing reports of fatal Powassan infections in the northeast (one very recently in Maine). So, scientists and practitioners and the public need to be alert to possible human cases of Bourbon Virus disease in New York State and elsewhere. Even more reason for personal protection against tick attachments when out of doors in 'tick-territory'. Might merit mention of these considerations within the mss. (if authors deem appropriate) - such as, for example use of permethrin-impregnated boots and clothing when in tick habitat.
Eisen L, Dolan MC. Evidence for Personal Protective Measures to Reduce Human Contact With Blacklegged Ticks and for Environmentally Based Control Methods to Suppress Host-Seeking Blacklegged Ticks and Reduce Infection with Lyme Disease Spirochetes in Tick Vectors and Rodent Reservoirs. J Med Entomol. 2016 Sep 1;53(5):1063-1092. doi: 10.1093/jme/tjw103. PMID: 27439616; PMCID: PMC5788731.
English language is fine. One typo: iotor instead of lotor
Round 2
Reviewer 1 Report
The revised manuscript has been substantially improved and multiple points have been clarified. No additional comments.